# Alkali-Treated Titanium Coated with a Polyurethane, Magnesium and Hydroxyapatite Composite for Bone Tissue Engineering

**DOI:** 10.3390/nano11051129

**Published:** 2021-04-27

**Authors:** Mahmoud Agour, Abdalla Abdal-hay, Mohamed K. Hassan, Michal Bartnikowski, Sašo Ivanovski

**Affiliations:** 1Department of Production Engineering and Design, Faculty of Engineering, Minia University, Minia 61112, Egypt; agour.mahmoud@gmail.com (M.A.); mkibrahiem@uqu.edu.sa (M.K.H.); 2Centre for Orofacial Regeneration, Reconstruction and Rehabilitation (COR3), School of Dentistry, Herston Campus, The University of Queensland, 288 Herston Road, Herston, QLD 4006, Australia; michal.bartnikowski@uq.edu.au; 3Department of Engineering Materials and Mechanical Design, Faculty of Engineering, South Valley University, Qena 83523, Egypt; 4Department of Mechanical Engineering, College of Engineering, Umm Al-Qura University (UQU), Mecca 24381, Saudi Arabia

**Keywords:** magnesium and its alloys, hydroxyapatite, surface modifications, titanium implants, corrosion analysis, biocompatibility, bioactivity

## Abstract

The aim of this study was to form a functional layer on the surface of titanium (Ti) implants to enhance their bioactivity. Layers of polyurethane (PU), containing hydroxyapatite (HAp) nanoparticles (NPs) and magnesium (Mg) particles, were deposited on alkali-treated Ti surfaces using a cost-effective dip-coating approach. The coatings were assessed in terms of morphology, chemical composition, adhesion strength, interfacial bonding, and thermal properties. Additionally, cell response to the variably coated Ti substrates was investigated using MC3T3-E1 osteoblast-like cells, including assessment of cell adhesion, cell proliferation, and osteogenic activity through an alkaline phosphatase (ALP) assay. The results showed that the incorporation of HAp NPs enhanced the interfacial bonding between the coating and the alkali-treated Ti surface. Furthermore, the presence of Mg and HAp particles enhanced the surface charge properties as well as cell attachment, proliferation, and differentiation. Our results suggest that the deposition of a bioactive composite layer containing Mg and HAp particles on Ti implants may have the potential to induce bone formation.

## 1. Introduction

Titanium (Ti) and its alloys are extensively used in orthopedic implants due to their favorable corrosion resistance, biocompatibility, machinability, and load bearing capability [1,2,3]. However, because of the bioinert nature of titanium, the osseointegration of Ti implants is relatively slow and can be compromised in sites of limited bone quality and quantity. Various methods of surface modification of Ti implants using physical, chemical, and biochemical treatment methods have been previously established [1,2,4,5], with the aim to enhance osseointegration of the Ti implants and promote their local bioactivity. Previous studies have shown that good surface wettability enhances the attachment, proliferation, and differentiation of osteoblasts and their precursors, leading to enhanced bone healing [4,6,7,8,9,10,11]. Additionally, surface chemistry plays an important role in the early stages of bone formation, with the greatest benefits derived from osteoconductive materials, such as hydroxyapatite HAp (Ca_10_(PO_4_)_6_(OH)_2_). HAp has been widely used for bone tissue engineering due to its excellent biocompatibility, bioactivity (chemical bonding ability with natural bone), slow in situ biodegradability, high osteoconductivity, osteoinductive potential, non-toxicity, and immunomodulatory properties [12,13]. However, HAp coatings are often problematic due to the low bonding strength of pure HAp coatings and Ti substrates [1,10,14,15,16,17,18,19,20,21]. The difference in the thermal expansion coefficients of Ti and HAp can cause delamination of the coating, leading to the failure of implants clinically.

Recently, magnesium (Mg) alloys or Mg coatings on metallic substrates were introduced in medical applications [1,22,23]. Our recent publication [4] showed that the incorporation of Mg particles into a biodegradable polymer not only enhanced the surface wettability, but also improved cell attachment and proliferation. We thus speculated that a composite polymer film composed of biodegradable metallic Mg particles and HAp NPs, successfully deposited on alkaline-treated Ti surfaces using a simple and cost-effective dip-coating method, is an efficient method to enhance the surface biocompatibility of Ti implants and subsequently improve osseointegration. The incorporation of HAp NPs within the polymer matrix may be capable of enhancing the interfacial strength between the alkali-treated Ti substrate and the polymer matrix by promoting increased hydrogen bonding and inducing superior adhesion. Previous studies have verified the strong interfacial strength between sodium titanate hydrogel layers formed on alkali-treated Ti and HAp particles [24,25,26]. In addition, HAp NPs present within the polymer matrix can become entrapped in the microporosity of the alkali-treated Ti surface, thus establishing mechanical interlocking with the Ti substrate. Furthermore, the incorporation of Mg particles in a biodegradable polymer matrix is advantageous as the release of acidic byproducts of polymer degradation can be compensated through the increase in pH facilitated by the dissolution of Mg [27,28,29,30]. Barrère et al. [31] demonstrated that the existence of Mg in an apatite (a phase of HAp) [13] coating of a titanium surface strengthened the adherence of the coating through the Mg ions, promoting further apatite precipitation.

Therefore, the aim of the present study is to incorporate both Mg metal and HAp NPs into a biodegradable polymer matrix and subsequently dip-coat a thin layer of this composite onto Ti substrates to enhance their bioactivity. Herein, polyurethane polymer (Pu) was utilized as a polymer matrix because it has a moderate degradation rate, good biocompatibility and exhibits a natural self-healing property [32]. In the present study, 5 wt.% synthesized HAp NPs [33] and 10 wt.% Mg microparticles [4,12] were incorporated into a PU polymer matrix before deposition through dip-coating onto alkali-treated Ti substrates. The morphological properties, chemical composition, surface roughness, wettability, adhesion strength, interfacial bonding, and thermal properties of the composite coating layer were investigated. Furthermore, MC3T3-E1 osteoblast-like cell adhesion, proliferation, and alkaline phosphatase (ALP) activity was studied in response to coated and uncoated Ti substrates.

## 2. Materials and Methods

The present study used a commercially pure titanium (c.p. Ti) sheet (William Gregor Ltd., London, UK), which was cut in a square shape with dimensions of 12 × 12 × 2 mm^3^. The sample preparation included grinding, polishing, and cleaning, according to previous publications [10,34,35,36,37,38]. Untreated samples were then etched in H_2_SO_4_:HCl:H_2_O = 1:1:1 volume ratio at 60 °C for 1 h to increase the surface roughness [39]. The hydrophilicity was enhanced by subjecting the etched samples to alkaline treatment using the same procedures as described elsewhere [35,36,39]. Briefly, the samples were immersed in 100 mL of an aqueous solution of 5 M NaOH at 60 °C for one day. Before further analysis, the samples were rained three times in distilled water and dried at 50 °C for 24 h at atmospheric conditions [10]. A HAp nanopowder was prepared using a wet chemical procedure, as described previously [4,12,33,40]. Further details on the preparation process of HAp is shown in the Appendix A. Scanning electron microscope (SEM) image and X-ray Diffraction (XRD) profiles of the synthesized HAp powder are shown in Figure 1a,b. The average diameter of the HAp particles was 120 ± 30 nm.

Coating process: Polymer solution containing 15 g of PU pellets (54,600 Da molecular weight, Estane skythane X595A-11, Lubrizol Advanced Material, Inc., Co., Ltd., Seoul, Korea) was dissolved in 100 mL dimethylformamide (DMF, Loba Chemie PVT. LTD., Mumbai, India). Mg powder with a purity of 99.9% (Kt, Sigma-Aldrich, St. Louis, MO, USA) and particle size range from 40 to 140 µm (SEM morphology and the XRD profile of the as-received Mg particles are shown in Figure 1c,d) was added into the PU solution at 10 wt.% (based on the PU weight contribution). Furthermore, 5 wt.% of HAp NPs, again based on the PU amount, was added into the Mg-particles/PU suspension solution.

Prior to the dip-coating process, all treated and untreated samples were pre-heated over a heating plate at 200 °C for 10 min to remove entrapped air and moisture from the surface. After this preheating process, the samples were immersed in the prepared solutions for about 30 s to facilitate wettability on the surface. Both dipping and withdrawing were performed at the same speed (2.0 mm·s^−1^) by a dip-coating machine (EF-5100, E-Flex, Bucheon, Korea), as displayed in Figure 2. The coated sheet was then hung in a vacuum oven (10 mbar) at 40 °C for 12 h. The drying temperature was selected based on the glass transition temperature of PU [41,42]. Due to the vertical orientation of the hung sample, there was an outflow of solution from the substrate that could affect the thickness of the coating film. However, previous studies have shown that the coating uniformity is improved when the sample is dried in a vertical rather than horizontal orientation [43,44,45]. Four groups of samples (S0–S3) were prepared and investigated. Table 1 describes the characteristics of the four groups. Characterization and cell culture experiment are shown in the Appendix A for further details.

## 3. Results and Discussion

The XRD curves of the surface of Ti treated with 5 M NaOH solution at 60 °C for 24 h are displayed in Figure 3. A small peak with a broad signal on the 2-theta axis at around 24° was observed. For the S0 group, outstanding peaks due to the α-phase of Ti always existed in the diffraction patterns [46]. Broad, small, and low intensity peaks were detected in the XRD profiles around the two-θ angle of 58°, which was due to the presence of an amorphous sodium titanate hydrogel layer with an atomic ratio similar to the Na_2_TiO_3_ phase formed by NaOH treatment, as illustrated in previous studies, including ours [33,46,47]. This is likely due to the significant chemical reaction that occurred after the NaOH treatment of the commercially pure Ti surface. However, a previous study [46] did not show the formation of a Na_2_TiO_3_ gel layer on the surface of Ti, which is in contrast with the present study. This further verifies the well-optimized parameters for NaOH treatment of Ti samples in the present work. The morphological properties and high surface area of the Na_2_TiO_3_ formed layer (as shown in Figure 4a) are the reason for the obtaining improvement in adhesion of the subsequent PU polymer coating, which will be discussed later. SEM (Figure 4b) and XRD (Figure 3) observations suggested the presence of a new layer of sodium titanate on the surface of S0, which is likely associated with enhanced adhesion performance of the coated films [10]. Figure 5a,b shows the Mg microparticle and HAp nanoparticle distribution within the PU matrix.

The phase composition of the PU film coated on the untreated and treated Ti surfaces was also investigated using the XRD technique. From the obtained results, it was found that the formation of weak, broad peaks refer to the polymeric phase profile on the Ti surface (see the left-hand side of XRD curves in Figure 3). In addition, the XRD of the composite coated films (S2, S3) showed weak peaks at approximately 34°, 36°, 48°, 57°, indicating the deposition of magnesium particles, and at about 32°, indicating the deposition of HAp particles [30]. SEM images (Figure 4) showed that the S2 group had a smooth and dense morphology, while the S3 group had well-developed upright (columnar) morphological structures, created on the alkaline Ti surface. The surface composition of the coated samples was analyzed by EDS, as depicted in Figure 4d, f. From the EDS analysis, it was confirmed that the Mg (Figure 4d) and HAp particles (Figure 4f) were successfully doped into the PU polymer coating layers. The thickness of the measured coating layer ranged from 8.14 to 9.74 µm, as shown in Figure 6.

The thermal stability of the plain PU coating and the composite coating was investigated using thermogravimetric analysis (TGA), as shown in Figure 6. It appeared that the addition of the Mg particles (Figure 7) improved the thermal stability of the PU as compared to the plain PU (Figure 6, S1). Additionally, the TGA thermogram indicated a similar single step degradation pattern for both S2 and S3. The char yield was higher for S3, which was due to the presence of the inorganic component of HAp. The glass transition temperature for the S3 group, at 50 °C, was marginally higher than that of the S2 group, potentially due to greater secondary interactions and the presence of an additional metal salt. This increase can reasonably be attributed to the coordinated interaction between oxygen and metals/metal ions. The melting point, around 167 °C, appeared to be influenced by the diol component or due to an ordered urea hydrogen bond formation. This may occur due to formation of amine in the presence of residual water/moisture. Finally, the peak around 320 °C indicated degradation. The improvement in degradation temperature in the composite groups is due to the excellent thermal stability of the Mg microparticles capsulated within the PU polymer matrix. In addition, the differential scanning calorimetry (DSC) scans (Figure 7) illustrate a slight shifting of the endothermic peaks and heating enthalpies of the coating groups, which is correlated with the melting of PU hard segments [48]. This might suggest typical interactions between the crystalline regions of PU and the Mg particles that enhance the stabilization of the dynamic thermal properties of the composite films.

It has been widely reported that coatings can easily delaminate due to a weak adhesion strength, significantly affecting the performance and stability of such coated biomedical devices [49]. In medical device development, adhesion stability between the metal substrate and coating is, therefore, vital to the successful use of implant. The degree of adhesion between the polymer layer and Ti substrate depends intimately on the stability and subsequently longevity of coatings. In the present study, the crosscut (known as tape) adhesion test was conducted to investigate the samples’ coating adhesion quality. Table 2 demonstrates the mean of the adhesion performance measurements. Interestingly, as expected, the S2 and S3 groups showed superior adhesion performance (5B, Table 2). It is assumed that the increased surface hydrophilicity (wettability) and the morphological changes caused by the alkali treatment improved the interfacial strength between the PU molecules and the treated Ti substrates.

The degree of surface roughness and charge (wettability) directly influence coating adhesion, with increased surface polarity and roughness resulting in increased reactivity between the substrate and coating. For example, we observed that very fine needle-like hairs were produced on the alkali-treated Ti (inset of Figure 1a). The high surface area of these features appeared to directly improve the adhesion of the deposited PU film [10]. The incorporation of HAp NPs and Mg microparticles into the PU film did not show negative effects on the adhesion performance of the deposited composite film, but instead also improved the adhesion strength.

It is well-known that the adhesion properties between a polymer coating and a metallic surface involves either physical or chemical bonding. In this case, chemical bonding involves a chemical interaction between the deposited PU molecules and species on the alkali-treated Ti surface. Indeed, the surface polarity of the Na_2_TiO_3_ gel layer formed on the Ti is higher than the untreated Ti surfaces, where the surface comprises an inert amorphous titanium dioxide film. The conversion of the Na_2_TiO_3_ gel layer to titanium-hydroxide (Ti-OH) after contact with an aqueous environment (such as water vapor) has been reported in our previous work [39], showing that this type of reaction is thermodynamically reasonable. The presence of OH on a treated Ti substrate may confer interesting properties to the Ti surface. For instance, the alkali-treated Ti surface may conduct as a base when in contact with a PU film. It is therefore suggested that the PU film, which has an acidic group, might strongly interact with the Na_2_TiO_3_ gel layer. These properties make this treatment a very attractive method to enhance the performance of polymer coatings. In fact, this kind of interaction may directly enhance the adhesion strength, as it has been illustrated that electrostatic force develops at the interface between materials that possess different electronic band structures [50]. Thus, the adhesion strength/bonding is strong in the case of the deposition of PU film on the active treated Ti surface.

This phenomenon can be discussed in light of the large number of free ends on the chain of PU, as it has been well discussed in our previous report as well [4], which provide a large number of free carboxyl groups for electrostatic interactions with the alkali-treated titanium surface, as shown in Figure 2. To support our hypothesis, Fourier-transform infrared spectroscopy (FTIR) analyses were used to investigate the molecular level interaction between the film coatings and substrates, as shown in Figure 8. A PO_4_^3−^ peak (566 cm^−1^) was present in the S3 group, similar to the one present in HAp. As is the case with biological HAp, peaks for carbonate (1411 and 1136 cm^−1^) were also present in the S3 group sample. The wide peak around 3361 cm^−1^ for the S1 and S2 groups were due to the presence of both hydrogen bonds and a free –NH functional group. After the addition of HAp, there was an increase in absorption due to the H-bonded –NH. The peak around 3500 cm^−1^ indicated the presence of –OH groups, which remained broad due to hydrogen bonding. The peak around 2927 cm^−1^ indicated –CH stretching. The sharp peak around 1136 cm^−1^ indicated the presence of the free –C=O group of S0, which reduced upon the addition of Mg (S1-Mg). When HAp was added, the peak again increased, indicating a shifting of interaction from Mg (with –C=O), to Mg with HAp. The peak around 1100 cm^−1^ indicated the presence of an ether linkage, while a peak for Mg(OH)_2_ was also present at 510 cm^−1^ in both S2 and S3 group samples [51].

The evaluation of cell/biomaterial interactions through the assessment of in vitro cytotoxicity test is one of the most fundamental initial tests for developed biomaterials [52]. Furthermore, the successful attachment of cells is an important part of this process and can strongly affect the subsequent cellular and tissue response. Figure 9 shows the cell morphology after three and five days of cell culture. The obtained images show the differential interaction of the cells with the different surfaces (some focusing issues arose from partial film detachment during the fixing, staining and dehydration processes).

For the S0 group, only a few, after three days of culture and after five days poorly spread cells were observed, no additional changes in cell morphology were noticed (Figure 9). The coating remained intact throughout the culture period. In the S2 and S3 groups, there were similarly no defects, detachments, or other degradations in the coating after 5 days of culture (Figure 9b–f), and a clear cellular spreading was observed, particularly in group S3. These results suggest that the S3 group had the most cytocompatible characteristics among the tested groups (Figure 9e,f). The S0 group possessed a higher surface roughness (1.110 ± 0.015 µm) than the S2 group (0.611 ± 0.020 µm), which would typically indicate that it should have superior performance in terms of cell migration, spreading. However, the S2 group surface also possessed a lower stiffness than the S1 surface, and indeed previous studies [53] showed that cell migration and focal adhesion are regulated by substrate flexibility. A study by Pelham and colleagues [54] found that cell spreading/migration could be guided by surface topography and physical interactions between the cell and the materials’ substrate. Their study concluded that changes in tissue rigidity and flexibility could play an important role in controlling cell spreading and migration. Additionally, cell spreading, and focal adhesions were affected by a change in mechanical properties of the extracellular matrix (ECM), which influenced cytoskeletal stiffness in vitro [52]. Ti-based implants possess a high stiffness that is responsible for the commonly problematic stress shielding phenomenon (i.e., there is a poor integration of the Ti implant with surrounding host tissues). Overall, it appears that the PU coating of alkali-treated Ti surfaces may regulate the adhesion and interactions of osteoblastic cells and the substrate.

Results of the MTT-assay confirmed that cell proliferation occurred over the 3 and 5 days of cell culture (Figure 10a). On day 3, it was observed that the S0 samples showed less cell proliferation than the S2 and S3 samples. After 5 d of culture, all groups showed a significant increase, and maintained over 80% viability. It has previously been reported that the formation of sodium titanate on the surface of Ti implants that are implanted in the body may have a harmful effect on cellular response because of the excessive release of sodium ions and the formation of narrow pore spaces [55]. We also observed that the Na^+^ ions released from the alkali-treated Ti surfaces had a negative effect on cell proliferation, despite the surface having an increased hydrophilicity [2]. However, the biocompatibility of the material was improved after coating, particularly with the S3 group samples, which performed the best of all of the groups.

Alkaline phosphate (ALP) is an early marker of osteoblast differentiation. We observed a clear increase in ALP activity after 8 and 14 d of culture, with ALP activity showing a significant increase in group S3 by day 14 (Figure 10b). Overall, the best outcomes in terms of cell proliferation and differentiation were observed in group S3. This observation indicates that there is a positive correlation between Mg^2+^ ion release into the media and an increase in osteogenic response.

The effect of Mg^2+^ ions on bone growth and bone formation was recently investigated [56]. The studies of the role of Mg^2+^ ions in accelerating bone formation have showed that the ions not only enhance bone adhesion and bone healing but also aid in the regulation and acceleration formation of bone marrow cells through the enhancement of BMP-receptor recognition, Smad signaling pathways, and/or the upregulation of neuronal calcitonin gene-related polypeptide (CGRP) [57,58]. The stimulation of bone formation by Mg^2+^ ions has been widely reported including ours, primarily through enhancing the activity of osteoblast, including adhesion/attachment, growth, and differentiation of these cells [30]. Mg^2+^ ions are actively involved in the process minerals formation to control bone formation and resorption [23]. Accordingly, the hypothesis that the S3 samples would increase MC3T3-E1 proliferation and differentiation compared with S0 samples, is accepted. Finally, it appears that coatings of both plain PU and PU doped with metallic Mg particles in a thin film deposited on a Ti implant are safe and effective in an in vitro cell culture model. Using a HAp/Mg/PU film coating on the Ti implants also further increased the level of osteoblast cell proliferation and differentiation, benefiting from the synergistic effects of both Mg and HAp. We have planned future work to investigate the microenvironment of the coated implants more comprehensively, and furthermore plan to conduct in vivo studies where we evaluate the coating durability, and the osteogenesis, osteoinduction, and osseointegration of these implants.

## 4. Conclusions

In this study, alkali-treated Ti substrates were coated with thin films of PU, PU with Mg particles, and PU with Mg and hydroxyapatite (HAp) particles using a dip-coating technology. The coatings were stable and did not delaminate from the implant surfaces. The use of dip-coating technology allowed us to create novel composites, introducing biologically compatible metal particles (Mg) and HAp onto alkali-treated Ti in a short time using a facile process. The morphology of the modified Ti surface interacted very favorably with the HAp NPs, increasing the adhesive strength between the substrate and the coating. Additionally, the surface charge (wettability) of the HAp NPs appeared to improve the interfacial strength by interacting with the sodium titanate gel layer formed on the treated Ti substrates. The HAp/Mg/PU group achieved the most positive effects in terms of enhancing preosteoblast adhesion, proliferation, and differentiation, which can be attributed to the bioactivity and biocompatibility of both the HAp particles and Mg^2+^ ions released from the metal particles. Overall, our results suggest that HAp/Mg/PU coated alkali-treated Ti is highly suitable as a biodegradable and bioactive orthopedic implant material.

## Figures and Tables

**Figure 1 nanomaterials-11-01129-f001:**
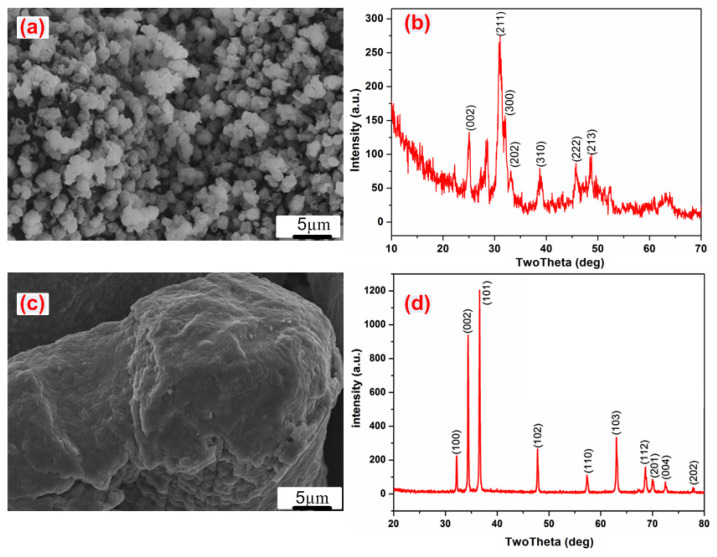
Scanning electron microscopy (SEM) images and X-ray diffraction (XRD) patterns of: (**a**,**b**) HAp, (**c**,**d**) Mg powders.

**Figure 2 nanomaterials-11-01129-f002:**
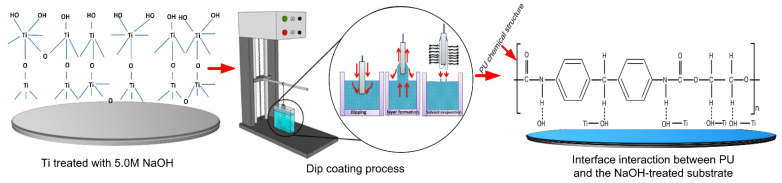
The dip-coating process and the considerable electrostatic intermolecular interaction between the polymer coating and the Ti substrate at the PU polymer side chain. Dotted line indicates electrostatic interaction.

**Figure 3 nanomaterials-11-01129-f003:**
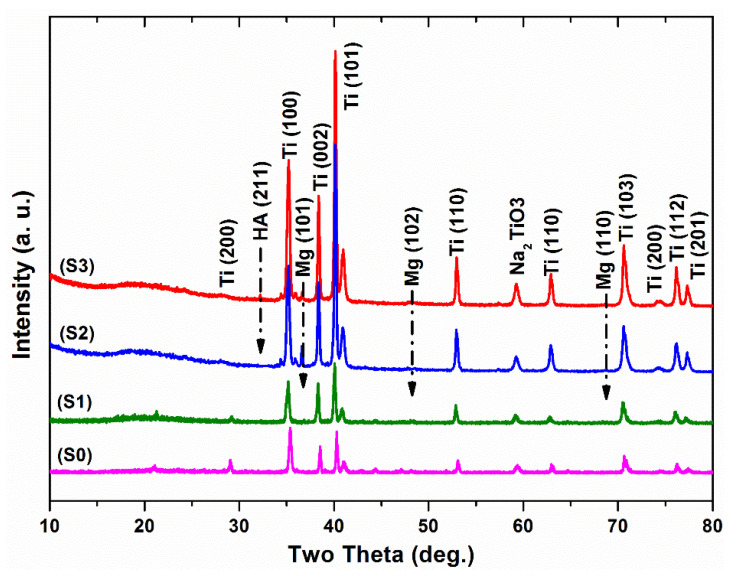
X-ray diffraction profiles of the coated samples. The formation of Na_2_TiO_3_ gel layer at two-theta with an angle of 58°, incorporation of Mg microparticles and HAp NPs within PU film at 2-theta, 36°, 48°, 57° for Mg particles and 32° for HAp NPs are illustrated by dashed vertical Dotted marks over the XRD profiles.

**Figure 4 nanomaterials-11-01129-f004:**
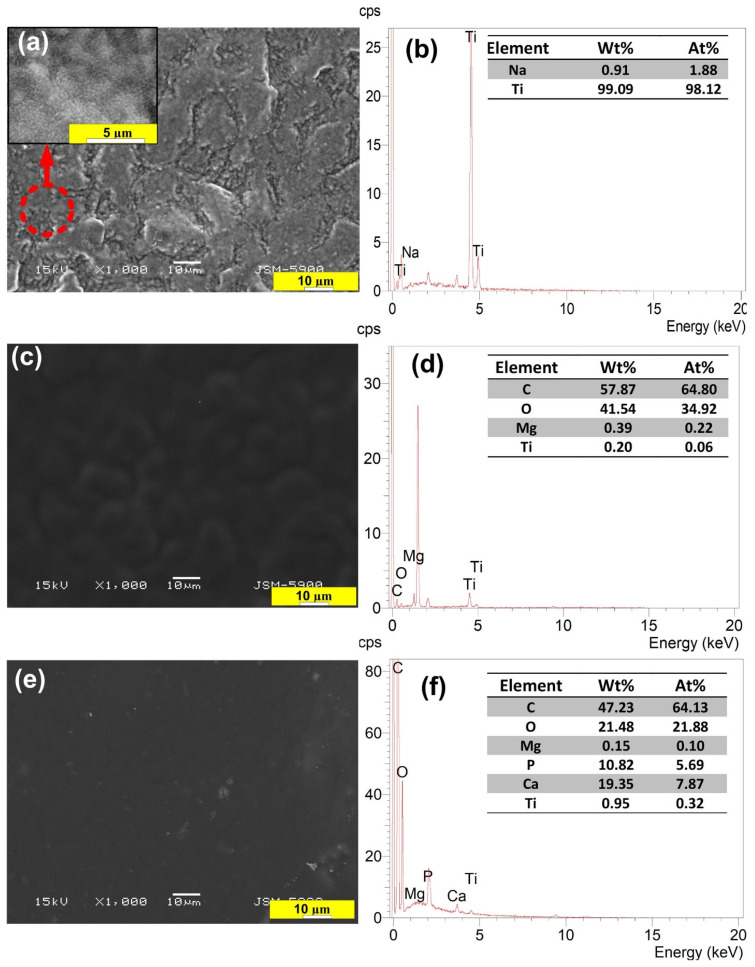
SEM images and their corresponding EDS profiles of S0 (**a**,**b**), S2 (**c**,**d**), and S3 (**e**,**f**) samples. The analysis of the corresponding SEM image are presented as insets of corresponding panel.

**Figure 5 nanomaterials-11-01129-f005:**
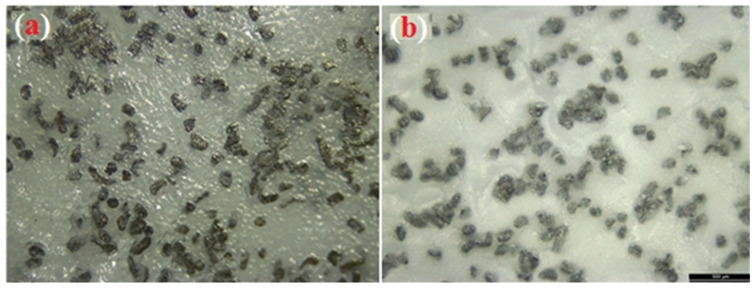
Optical microscopy images showing the distribution of Mg particles (**a**) and Mg-particles/HAp (**b**) throughout the PU layer.

**Figure 6 nanomaterials-11-01129-f006:**
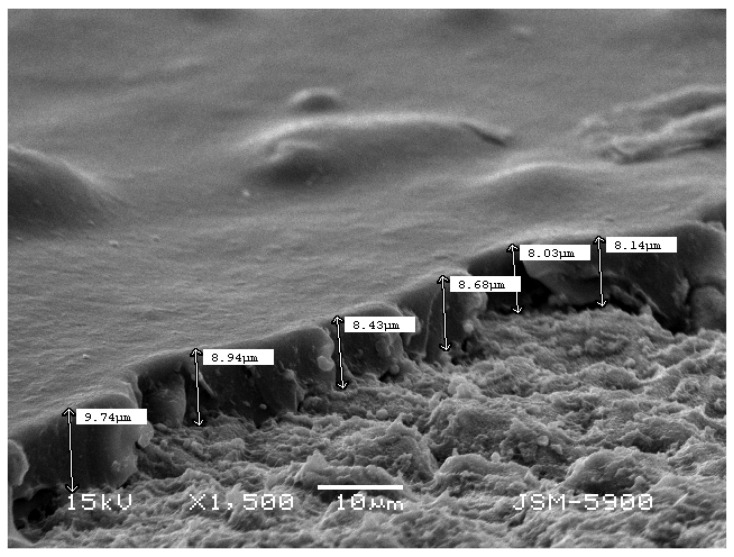
SEM of the cross-section of the composite layer coating on the Ti surfaces, indicating the coating thickness.

**Figure 7 nanomaterials-11-01129-f007:**
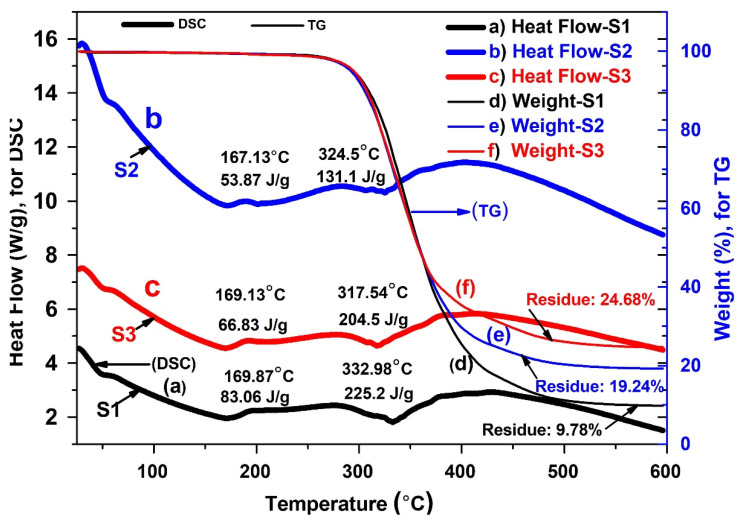
Simultaneous thermogravimetric analysis/differential scanning calorimetry (TGA/DSC) of the three experimental groups comprising plain: (**a**–**c**) heat flow and (**d**–**f**) residual weight of plain and composite layers coated on Ti substrates.

**Figure 8 nanomaterials-11-01129-f008:**
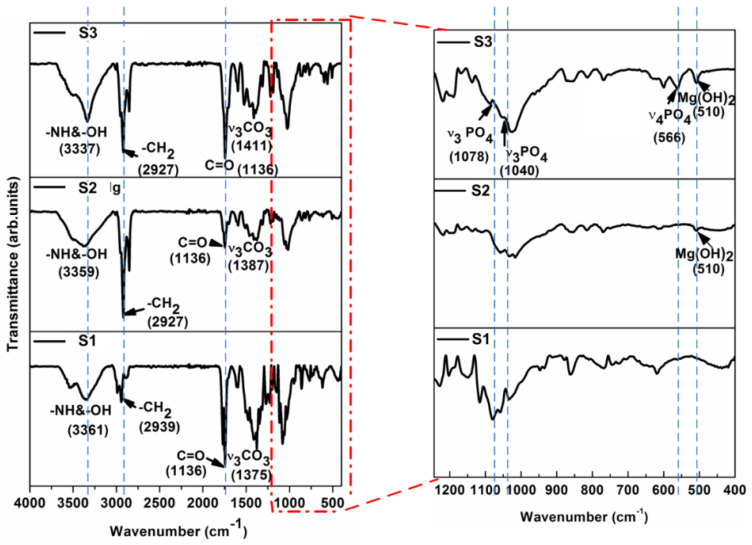
FT-IR spectra in transmission mode of a typical sample surface, comprising Ti coated with PU with and without Mg and HAp particles, after a 5 M NaOH chemical treatment at 60 °C for 24 h of the Ti samples.

**Figure 9 nanomaterials-11-01129-f009:**
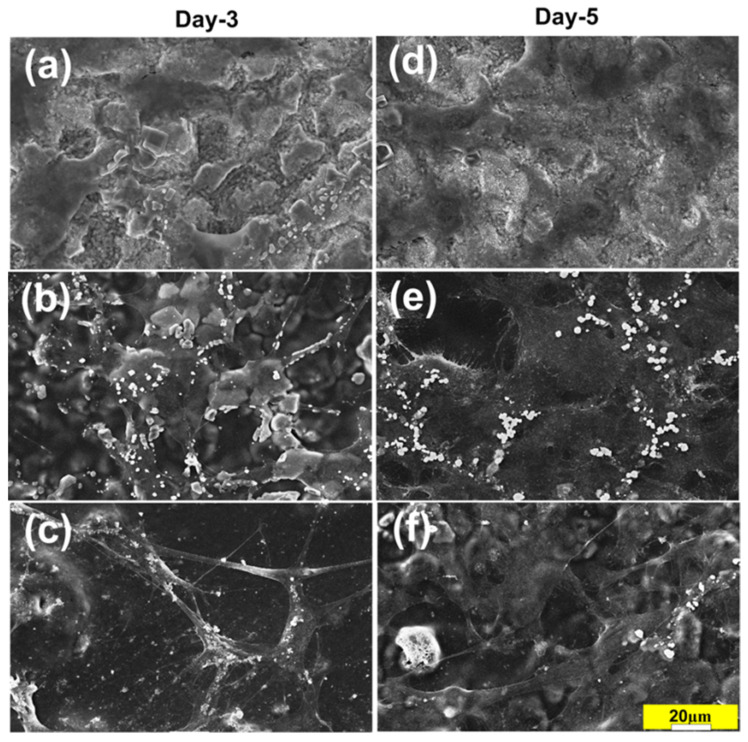
SEM images of MC3T3 osteoblastic cells lines cultured for three and five days on the treated and treated Ti surfaces of S0 (**a**,**d**), S2 (**b**,**e**) and S3 group (**c**,**f**). Scale bar 20 µm.

**Figure 10 nanomaterials-11-01129-f010:**
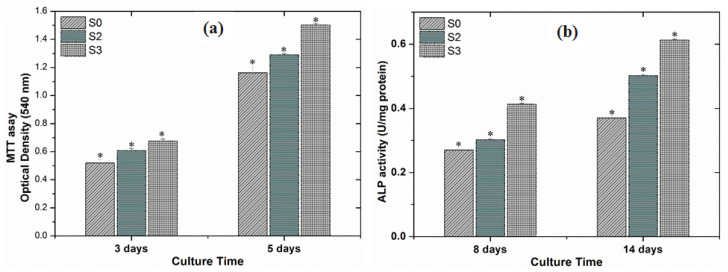
(**a**) Cell proliferation assay after 3 and 5 d of culture and (**b**) Alkaline phosphate activity (ALP) (U/mg protein) after 8 and 14 d of culture; cells were seeded on coated alkaline treated Ti samples (n = 3, * indicates *p* ≤ 0.05).

**Table 1 nanomaterials-11-01129-t001:** Designation of sample groups used in this study.

Sample Designation	Description
S0	Alkaline-treated Ti
S1	Plain PU film coated on alkaline-treated Ti
S2	Mg-particle/PU composite film coated on alkaline-treated Ti
S3	Mg- HAp/PU composite film coated on alkaline-treated Ti

**Table 2 nanomaterials-11-01129-t002:** Cross-cult adhesion test results of Mg particles/PU and HAp-Mg particles/PU composite coatings formed on alkali-treated Ti (assessed by ASTM (D3359, 2010), a standard protocol).

Samples	Rating of Extent of Adhesion as per ASTM (D3359, 2010)	Percent Area Removed *	Surface Appearance after Adhesion Testing
S2	5B	0% (none)	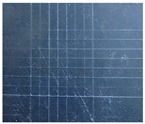
S3	5B	0% (none)	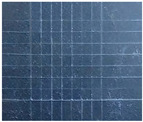

* Percent area removed was determined based on the classification of adhesion test results of D3359 ASTM standard for measuring adhesion by tape test, according to the following criteria: 5B: The edges of the cuts are completely smooth; none of the squares of the lattice are detached.

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
