# Peer review of "Alkali-Treated Titanium Coated with a Polyurethane, Magnesium and Hydroxyapatite Composite for Bone Tissue Engineering"

_nanomaterials, 2021, doi:10.3390/nano11051129_

Round 1

Reviewer 1 Report

Ms. ID: nanomaterials-1163231

Title: Alkali-Treated Titanium Coated with a Polyurethane, Magnesium and Hydroxyapatite Composite for Bone Tissue Engineering

Reviewer comments

Overall it is an interesting manuscript that addresses the issue of the surface modification of titanium for medical use. The goal of the study is to use a double surface modification of titanium. The first step is to enhance surface porosity and chemistry via alkaline treatment, and the second step is an original method developed by the authors that encompass the application of hydroxyapatite and particles embedded in polyurethane matrix. Based on the presented results, the method seems effective. Of potential use would be a demonstration of long-term stability of the synthesized materials, either using in vitro or in vivo tests in an animal model.

Specific comments

Line 87: “This section is shown in supporting information for more details.” It is a rather odd choice, and it is not hard to find a justification for it. The materials and methods section is a cornerstone of every scientific publication, and there is no way around it. Other parts should be adjusted (shortened) if the length (word count) is an issue.

Line 96: “are displayed in Figure 3.” It Looks like the figures are referenced in the text, not in order of their appearance (unless it is permissible).

Line 183: “Figure 7. Simultaneous thermogravimetric analysis/differential scanning calorimetry (TGA/DSC)”. This figure should be split into two parts: 1) DSC and 2) TGA, or at least the labeling and line color/style changed to avoid confusion among readers less familiar with both techniques.

Line 222: “Figure 8. FT-IR spectra in transmission mode of a typical sample surface,”. This figure could be widened to make spectra more legible.

Line 258: “Figure 9. SEM images of MC3T3 osteoblastic cells lines…” Why are there no results demonstrating the viability of the cells, which is a standard practice in evaluating the cytotoxicity of the new biomaterials?

Line 297: “Figure 10. (A) Cell proliferation assay after 3 and 5 days of culture…”. Usually, this kind of test is supplemented with the results of the proliferation on a standard and well-characterized material. In many cases, researchers choose to use polystyrene which is used to make tissue culture plates (TCPs). Why was it not done in this work?

Author Response

We thank all of the reviewers for their detailed and helpful inputs. Please see below for address of specific comments. Our responses are highlighted in blue colour and italicised.

Thank you for considering our revised manuscript for possible publication in the journal ‘Nanomaterials
Please feel free to contact us if you need any further details.

Sincerely

Abdalla Abdal-hay &Saso Ivanovski, PhD

Reviewers' comments:

Reviewer 1#

Overall it is an interesting manuscript that addresses the issue of the surface modification of titanium for medical use. The goal of the study is to use a double surface modification of titanium. The first step is to enhance surface porosity and chemistry via alkaline treatment, and the second step is an original method developed by the authors that encompass the application of hydroxyapatite and particles embedded in polyurethane matrix. Based on the presented results, the method seems effective. Of potential use would be a demonstration of long-term stability of the synthesized materials, either using in vitro or in vivo tests in an animal model.

Response: Thank you.

Specific comments

Line 87: “This section is shown in supporting information for more details.” It is a rather odd choice, and it is not hard to find a justification for it. The materials and methods section is a cornerstone of every scientific publication, and there is no way around it. Other parts should be adjusted (shortened) if the length (word count) is an issue.

Response: we agree with the reviewer concern, hence, we have moved the essential part of the materials and methods to the main text

Line 96: “are displayed in Figure 3.” It Looks like the figures are referenced in the text, not in order of their appearance (unless it is permissible).

Response: this has been modified, and we followed the sequential order of the figures in the manuscript.

Line 183: “Figure 7. Simultaneous thermogravimetric analysis/differential scanning calorimetry (TGA/DSC)”. This figure should be split into two parts: 1) DSC and 2) TGA, or at least the labeling and line color/style changed to avoid confusion among readers less familiar with both techniques.

Response: The figure has been modified as requested (Fig. 7).

Line 222: “Figure 8. FT-IR spectra in transmission mode of a typical sample surface,”. This figure could be widened to make spectra more legible.

Response: a range of wavelengths from 400 to 1300 cm-1 are highlighted on the amin curve to further show the important peaks relevant to the study

Line 258: “Figure 9. SEM images of MC3T3 osteoblastic cells lines…” Why are there no results demonstrating the viability of the cells, which is a standard practice in evaluating the cytotoxicity of the new biomaterials?

Response: The cell viability of the proposed materials was widely investigated by several authors, including ours (see below attached references); hence in the present study we focused on investigating the metabolic activity using MTT test (reflects proliferation of viable cells), cell attachment and differentiation using ALP activity.

ine 297: “Figure 10. (A) Cell proliferation assay after 3 and 5 days of culture…”. Usually, this kind of test is supplemented with the results of the proliferation on a standard and well-characterized material. In many cases, researchers choose to use polystyrene which is used to make tissue culture plates (TCPs). Why was it not done in this work?

Response: the reviewer is correct -TCP is often used as a generic control. In our previous study, plain Ti disc without any surface modifications was used as an appropriate and relevant negative control group (European Polymer Journal 112, 555-568). In the current study, to further advance our line of investigation, alkali treated Ti samples were used as an appropriate negative control and compared with Ti disc coated with polymer composite coatings, to avoid repetition of data presented in our previous publication, European Polymer Journal 112, 555-568.

Reviewer 2 Report

The work is very well structured, is of interest to readers, is of superior quality. I recommend for publication. 

Author Response

We thank all of the reviewers for their detailed and helpful inputs. Please see below for address of specific comments. Our responses are highlighted in blue colour and italicised.

Thank you for considering our revised manuscript for possible publication in the journal ‘Nanomaterials
Please feel free to contact us if you need any further details.

Sincerely

Abdalla Abdal-hay &Saso Ivanovski, PhD

Reviewers' comments:

Reviewer 2#

The work is very well structured, is of interest to readers, is of superior quality. I recommend for publication.

Thank you for your kind words!

Reviewer 3 Report

The paper entitled ‘Alkali-Treated Titanium Coated with a Polyurethane, Magnesium and Hydroxyapatite Composite for Bone Tissue Engineering’ describes optimization of surface properties of titanium implants and brings evidence that composite coating containing hydroxyapatite nanoparticles and magnesium particles in polyurethane layer improves both mechanical properties of the coating and bioactivity. The methods are appropriate to the research described and the results are clearly presented.

Here are my comments and suggestions to be considered:

Major points:

  1. It is an article about modification of material surfaces. Materials and methods should be a key part of the text. Why it is almost completely a supplementary material?

          References in the manuscript include those in the supplementary part.

  1. Line 260 and Figure 10A refer to MTT assay, it does not match the text in the methods in the supplement. There is about CCK-8 and WST assay.

Minor points:

  1. Splitting words at the end of the line in the title ?
  2. Different order of the authors in the submission system, manuscript and supplementary file.

     Abdalla Abdal-hay, Mahmoud Agour, ...    or    Mahmoud Agour, Abdalla Abdal-hay,...

  1. Check the affiliations :

     Minia Unviversty ?

     Mechanical Engineering Dept., College of Engineering, Umm AlQura University, KSA,  ?

  1. line 42: Ca10(PO4)6(OH)2

5.Check the references, some are the same!

    2=48    46=47 43=44  10=24

     ref. 34: Abdal-ha, A.      Abdal-hay ?

  1. Table 1: alkali-treated alkaline-treated     It could be unified.
  2. Check the abbreviations: TKP (line 216) NPs (line 308)
  3. Spelling errors: line 154: th e Mg particles that enhance the stablization

                             line 243:  mifration

                             paragraph 225-231

Author Response

We thank all of the reviewers for their detailed and helpful inputs. Please see below for address of specific comments. Our responses are highlighted in blue colour and italicised.

Thank you for considering our revised manuscript for possible publication in the journal Nanomaterials
Please feel free to contact us if you need any further details.

Sincerely

Abdalla Abdal-hay &Saso Ivanovski, PhD

Reviewers' comments:

Reviewer 3#

The paper entitled ‘Alkali-Treated Titanium Coated with a Polyurethane, Magnesium and Hydroxyapatite Composite for Bone Tissue Engineering’ describes optimization of surface properties of titanium implants and brings evidence that composite coating containing hydroxyapatite nanoparticles and magnesium particles in polyurethane layer improves both mechanical properties of the coating and bioactivity. The methods are appropriate to the research described and the results are clearly presented.

Response: Thank you for the feedback.

Here are my comments and suggestions to be considered:

Major points:

  1. It is an article about modification of material surfaces. Materials and methods should be a key part of the text. Why it is almost completely a supplementary material?

          References in the manuscript include those in the supplementary part.

Response: We agree with the reviewer concern, hence we have moved the essential part of the materials and methods to the main text.

  1. Line 260 and Figure 10A refer to MTT assay, it does not match the text in the methods in the supplement. There is about CCK-8 and WST assay. For the determination of cellular attachment and cellular growth,
    MC3T3-E1 osteoblast cell lines were cultured with Ti samples.

Response: This section was corrected and modified.

Minor points:

  1. Splitting words at the end of the line in the title ?
  2. Different order of the authors in the submission system, manuscript and supplementary file.

     Check the affiliations :

     Minia Unviversty ?

     Mechanical Engineering Dept., College of Engineering, Umm AlQura University, KSA,  ?

In the two universities with email mkibrahiem@uqu.edu.sa

  1. line 42: Ca10(PO4)6(OH)2

 3.Check the references, some are the same!

    2=48    46=47 43=44  10=24

     ref. 34: Abdal-hay, A.      Abdal-hay ?

  1. Table 1: alkali-treated alkaline-treated     It could be unified.
  2. Check the abbreviations: TKP (line 216) NPs (line 308)

S1 (Line 216) NPs

  1. Spelling errors: line 154: the Mg particles that enhance the stabilization

                             line 243:  mifration

                             paragraph 225-231

Response: Thank you. All comments were addressed accordingly.